# Chemical Profiling of *Hedyosmum cumbalense* and *Hedyosmum spectabile* (Chloranthaceae) Essential Oils, and Their Antimicrobial, Antioxidant, and Anticholinesterase Properties

**DOI:** 10.3390/plants12010039

**Published:** 2022-12-22

**Authors:** Alisson Guerrero, Emilye Guerrero, Luis Cartuche, Nixon Cumbicus, Vladimir Morocho

**Affiliations:** 1Carrera de Bioquímica y Farmacia, Universidad Técnica Particular de Loja (UTPL), Loja 1101608, Ecuador; 2Departamento de Química, Universidad Técnica Particular de Loja (UTPL), Loja 1101608, Ecuador; 3Departamento de Ciencias Biológicas y Agropecuarias, Universidad Técnica Particular de Loja (UTPL), Loja 1101608, Ecuador

**Keywords:** *Hedyosmum cumbalense*, Hedyosmum spectabile, essential oils, anticholinesterase, sabinene

## Abstract

In Ecuador, *Hedyosmum cumbalense* and *Hedyosmum spectabile* are valued for their well-known aromatic characteristics and therapeutic benefits. In this study, fresh and dried leaves of these species were steam-distilled to obtain their essential oils (EOs) for chemical characterization and assessment of their antimicrobial, antioxidant, and anticholinesterase properties. Gas chromatography coupled to mass spectrometry (GC-MS) and a flame-ionized detector (GC-FID) with a nonpolar column was employed to determine the chemical composition, along with the enantioselective analysis. The antimicrobial activity was evaluated against three Gram-positive, two Gram-negative, and two sporulated fungi. The radical scavenging properties were evaluated by DPPH and ABTS assays. A total of 66 and 57 compounds were identified for *H. cumbalense* and *H. spectabile*, respectively. Two pairs of enantiomers for each species were also detected, with (1R,5R)-(+)-sabinene and (1S,5S)-(–)-sabinene found in both specimens. A poor effect against Gram-positive cocci was observed on *H. cumbalense* (MIC of 4000 µg/mL). Both oils displayed weak antifungal activities, exhibiting a MIC of 1000 µg/mL. *H. cumbalense* had a good scavenging effect assessed by the ABTS radical (SC_50_ = 96.02 ± 0.33 µg/mL). Both EOs showed a strong anticholinesterase potential with an IC_50_ value of 61.94 ± 1.04 µg/mL for *H. cumbalense* and 21.15 ± 1.03 µg/mL for *H. spectabile*.

## 1. Introduction

Chloranthaceae is a small family with around 70 species. It is among the oldest lineages of angiosperms. Plants belonging to the Chloranthaceae family can be easily identified by their jagged leaves and primitive odor. The presence of secretory cells in stems and leaves is also a typical feature [1]. The family is composed of species of four genera, *Ascarina*, *Chloranthus*, *Hedyosmum*, and *Sarcandra,* and they are found in tropical and subtropical regions of South America, East Asia, and the Pacific [2].

In the genus *Hedyosmum (Tafalla, Tafallaea, or Tavalla* genus, heterotypic synonyms), there are 48 species of tiny trees and shrubs and 45 of them have been taxonomically identified and considered as distinct species [3]. The word “*Hedyosmum*” is derived from the Greek words “Hedy-,” which means sweet, pleasant, fragrant, and “osme,” meaning smell [4]. The *Hedyosmum* genus is primarily found in tropical America, in low and high mountain forests like those of South America’s Andes. It is the Chloranthaceae genus with the widest distribution in America, primarily in Ecuador, Peru, Brazil, and central Bolivia [5].

The plants of the genus *Hedyosmum* are defined as medium-sized shrubs or trees with long-legged roots, segmented, knotty, and fleshy-looking twigs, and unisexual flowers [4]. When crushed, its leaves release an odor that is either aromatic or astringent. Their leaves are wrinkled and have serrated or serrulate borders. Several of the naturally occurring active compounds found in *Hedyosmum* species have been employed in folk medicine. The aerial components, including the leaves, bark, and fruits, have been employed for traditional remedies and ethnomedical practices [6].

The most frequent method of preparation usually involves the production of alcoholic beverages and infusions from the leaves. The economic importance of species of this genus lies in the fact that they are used as a source of food, medicine, firewood, and building materials. However, sedatives (therapy for pain brought on by a cold, rheumatic joint discomfort, and fever), aphrodisiacs, antidepressants, and remedies for treating stomach pain (digestive, antispasmodics, and stomach-soothing) are the most popular and traditional applications of *Hedyosmum* species [7].

In terms of chemical composition, Chloranthaceae plants are rich in sesquiterpenes of the eudesmane, lindenane, guaiane, germacrane, cadinane, and aromadendrane-type compounds. Sesquiterpenes, which are among the distinctive chemical elements of the Chloranthaceae family of plants, may be the cause of many antibacterial, antitumor, and other properties [8]. Sesquiterpenes and monoterpenes, particularly β-pinene and sabinene (which possess analgesic, anti-inflammatory, and antifungal properties), are the most prevalent compounds found in the essential oils of *Hedyosmum* species. Essential oils also contain germacrene D, pinocarvone, and α -phellandrene. Numerous investigations on their biological activities generally center on sesquiterpenes and sesterterpenes [7].

*Hedyosmum spectabile* Todzia., is one of the most recently recorded species within the genus and one of the most distinctive by its unusual leaf sheaths. According to Todzia [9], these plants are often small, aromatic trees or shrubs that grow 3–7 m tall, have prop roots, white wood that becomes orange when cut, purple stems, and persistent leaf sheaths around the nodes. When crushed, the leaves of *H. spectabile* emit a strong aroma that is reminiscent of anise, lemon, and turpentine. Because it is a new species, not much is known about its applications.

*Hedyosmum cumbalense* H. Karst is a tree or shrub that can grow up to 6 m in height, with fragrant, creamy yellow flowers. It is primarily found in Peru, Ecuador, and Colombia [10]. In the province of Morona Santiago, the plant is used to make tea and is commonly known as “chumat” in Kichwa and “guayusa” in Spanish. In Carchi, locals burn it to obtain charcoal which they may use as fuel. Additionally, in Imbabura and Carchi provinces, it is employed in carpentry as well as in the construction of walls and posts. Another customary usage also primarily serves as flavorings for human consumption [11].

Although the essential oils of the genus *Hedyosmum* have recently been the subject of substantial research due to their potential therapeutic properties, there are no reports on the chemical composition of the essential oil from leaves of *H. spectabile* and *H. cumbalense*. For this reason, this research aimed to determine the chemical composition, enantiomeric distribution, and antimicrobial, antioxidant, and anticholinesterase activity of the *H. cumbalense* and *H. spectabile* essential oils.

## 2. Results

The essential oils were obtained by steam-distillation from leaves of *Hedyosmum cumbalense* and *Hedyosmum spectabile*. Both oils were obtained as a translucent viscous liquid, with *H. cumbalense* samples being slightly yellow. The mean percentage yield of EOs was 0.15 ± 0.05% and 0.25 ± 0.05% for *H. cumbalense* and *H. spectabile*, respectively.

### 2.1. Chemical Composition of EO from Leaves of H. cumbalense

The essential oil of *H. cumbalense* was analyzed by gas chromatography-mass spectrometry (GC-MS) and gas chromatography-flame ionized detector (GC-FID) using nonpolar column DB-5 ms for the identification of volatile components. The results from the GC analysis of *H. cumbalense* are shown in Table 1. A total of 66 compounds were identified, representing 95.94% of the total chemical composition. The main chemical compound was sabinene (14.37 ± 1.64%), followed by isobornyl acetate (9.12 ± 1.11%), α-pinene (7.91 ± 2.10%), *β*-pinene (7.41 ± 1.46%) and linalool (4.94 ± 0.93%), accounting for more than 43.75% of the chemical composition. Other minor occurring compounds (≤5%) are thymol, methyl ether (4.75 ± 0.66%), citronellal (3.51 ± 0.37%), camphene (3.01 ± 0.72%), eugenol (2.55 ± 0.99%), terpinen-4-ol (2.43 ± 0.86%) and sylvestrene (2.07 ± 0.17%). The main chemical group in this specie was monoterpenes which constitute more than 61.59% of the compounds identified: oxygenated monoterpenes (37.79%) and hydrocarbonated monoterpenes (21.80%).

### 2.2. Chemical Composition of EO from Leaves of H. spectabile

The qualitative and quantitative analysis of the compounds of *H. spectabile* was carried out using gas chromatography coupled to MS and to FID detectors, with a nonpolar column DB-5 ms. Fifty-seven compounds were detected and quantified, which cover 92.37% of the Eos’ total composition. The data obtained from the GC analysis is shown in Table 2. cis-Muurola-4(14),5-diene (17.87 ± 2.79%) was the principal constituent in the EO, along with muurola-4,10(14)-dien-1-*β*-ol (6.55 ± 0.58%), aciphyllene (5.37 ± 0.27%), (E)-*β*-ocimene (5.35 ± 1.01%) and 1,3,8-*ρ*-menthatriene (4.99 ± 1.15%). The other main compounds were *β*-elemene (3.69 ± 0.87%), cubenol (2.77 ± 0.22%), *α*-copaene (2.48 ± 0.46%) and *δ*-amorphene (2.44 ± 0.46%). According to their chemical nature, sesquiterpenes were the dominant group of constituents in this species, which represent 71.30% of the compounds identified: hydrocarbonated sesquiterpenes (40.51%) and oxygenated sesquiterpenes (30.79%).

### 2.3. Enantioselective GC Analysis of the Essential Oils

The enantioselective analysis was determined using a GC column coated with 2,3-diacethyl-6-tert-butylsilyl-*β*-cyclodextrin as a chiral selector. In *H*. *spectabile*, linalool exhibited a high enantiomeric excess (e.e.) while sabinene was almost racemic, with only a small e.e. in favor of (1S,5S)-(–)-sabinene. For *H. cumbalense*, both (1S,5S)-(–)-sabinene and (R)-(−)-terpinen-4-o displayed a high enantiomeric excess. The results are shown in Table 3, in which the enantiomeric distribution, the linear retention indices, and the enantiomeric excess values are included.

### 2.4. Antimicrobial Activity of H. cumbalense and H. spectabile

The antibacterial activity of essential oil of *H. cumbalense* and *H. spectabile* leaves were assessed through the microdilution broth method. Ampicillin was used as a positive control for Gram-positive, Ciprofloxacin for Gram-negative, and Amphotericin B for yeasts and sporulated fungi. The results of the activity of the essential oils against human pathogenic microorganisms are displayed in Table 4, including the minimum inhibitory concentration (MIC) values and the microorganisms used (three Gram-negative bacteria, three Gram-positive bacteria, and two yeasts and sporulated fungi). *H. cumbalense* demonstrated weak or null activity against *Enterococcus faecium* and *Staphylococcus aureus* at a dose of 4000 µg/mL. For Gram-negative bacilli, neither of the essential oils showed antimicrobial activity. A higher effect was displayed on yeasts and sporulated fungus. For *Aspergillus niger*, both EO exhibited a MIC of 1000 µg/mL. Meanwhile, with *Candida albicans*, only *H. cumbalense* afforded a MIC of 1000 µg/mL.

### 2.5. Antioxidant Capacity

The essential oils of both species were tested for antioxidant activity. This was done by using ABTS and DPPH radicals. Trolox was used as a positive reference substance. The data obtained are depicted in Table 5.

### 2.6. Anticholinesterase Activity

The inhibitory effect of both essential oils is depicted in Figure 1. Calculated IC_50_ values for *H. spectabile* and *H. cumbalense* essential oils were 21.15 ± 1.03 and 60.91 ± 1.04 µg/mL, respectively. The anticholinesterase activity of the positive reference control, donepezil, exhibited an IC_50_ value of 12.40 ± 1.35 nM.

## 3. Discussion

The essential oils of *H. cumbalense* and *H. spectabile* exhibited a poor yield of 0.15 ± 0.05% and 0.25 ± 0.05%, respectively. *H. spectabile* is one of the most recent species registered in the genus, which explains the lack of information about its essential oil extraction. In the same way, although *H. cumbalense* is more recognizable among native people, there is still not enough knowledge about its EO. In general, the majority of *Hedyosmum* species presented a low yield of essential oil by hydrodistillation; however, this value is very variable and depends on several factors such as the part of the plant used, the preparation of the material (dried or fresh leaves), or the extraction time and method, including hydrodistillation, steam-distillation, etc. [12].

Based on the results, the chemical profile of both essential oils showed an unexpected variation being the main differences, the occurrence of sabinene, isobornyl acetate, α-pinene, β-pinene, linalool and thymol methyl ether with percentages higher than 5%, as the main constituents in *H. cumbalense*, while as in *H. spectabile*, the main occurring compounds with percentages higher than 5% are cis-muurola-4(14),5-diene, muurola-4,10(14)-dien-1-β-ol, aciphyllene, cis-*β*-ocimene and chrysanthenone.

Despite having different main constituents, some minor components are shared in different proportions. A total of 14 common compounds were identified, which are α-pinene (1.64 and 7.91%), sabinene (0.41 and 14.37%), *β*-pinene (0.25 and 7.41%), myrcene (0.24 and 0.56%), sylvestrene (0.29 and 2.07%), *trans*-*β*-ocimene (1.84 and 1.30%), *cis*-*β*-ocimene (5.35 and 0.64%), linalool (0.99 and 4.94%), chrysanthenone (0.68 and 0.22%), *E*-caryophyllene (1.04 and 0.19%), *δ*-amorphene (2.44 and 0.25%), germacrene B (0.94 and 0.23%), spathulenol (1.34 and 0.71%), and globulol (2.03 and 0.12%) for *H. spectabile* and *H. cumbalense*, respectively.

According to Radice et al., the above-mentioned compounds are recurrent constituents in many essential oils of the *Hedyosmum* species [7]. It is also indicated that terpenes are the primary chemical group present in this genus, which includes sesquiterpenoids and monoterpenes. These types of compounds represent another difference between the EOs. Based on the data obtained, the *H. spectabile* dominant group of compounds was sesquiterpenes (71.30%), which includes mainly hidrocarbonated sesquiterpenes. On the contrary, more than 50% of *H. cumbalense* essential oil was constituted by monoterpenes (61.59%), mostly by oxygenated monoterpenes.

The enantioselective GC-MS analysis showed the presence of two pairs of enantiomers for each species: (S)-(+)-terpinen-4-ol and (R)-(–)-terpinen-4-ol for *H. cumbalense*, (S)-(+)-linalool and (R)-(-)-linalool for *H. spectabile,* and (1R,5R)-(+)-sabinene and (1S,5S)-(–)-sabinene, in both specimens. It is common to find these chiral components in the *Hedyosmum* genus, as seen in *H. angustifolium* Ruiz & Pav. and *H. scabrum* Ruiz & Pav. [13], though the enantiomeric distribution of each component differs depending on the species.

All chiral compounds except sabinene found in *H. spectabile*, which has an e.e. of 4.25%, have significant levels of enantiomeric excess. The chiral chemicals found in *H. spectabile* make up 1.4% of the essential oil’s overall composition but are not among its major constituents. On the other hand, *H. cumbalense* chiral compounds account for 16.8% of its total oil content. This percentage is significant enough to raise the possibility that chiral chemicals may influence the biological activity observed. This is the first time either plant has undergone an enantioselective examination.

Regarding Gram-negative bacilli, none of the essential oils showed activity against the two strains of bacteria examined: *Escherichia coli* (O157:H7) and *Pseudomonas aeruginosa*. On the other hand, a weak effect against Gram-positive cocci was reported for the EO of *H. cumbalense*, which showed a MIC of 4000 µg/mL against *Enterococcus faecium* and *Staphylococcus aureus*. *H. spectabile* was inactive. According to the classification of the biological activity of essential oils proposed by Van Vuuren and Holl, D., the MIC values equal to or over 1000 µg/mL should be considered noteworthy of publication [14]. Contrary to the results of antibacterial activity, the effects obtained for yeasts and sporulated fungi were presented in both essential oils. For *Aspergillus niger*, both plants presented a MIC of 1000 µg/mL. However, for *Candida albicans,* the MIC value for *H. spectabile* was higher (2000 µg/mL).

Although *H. cumbalense* and *H. spectabile* have not previously been studied, the findings are comparable to those of other *Hedyosmum* species since it can also be observed that the antimicrobial activity is equally low in different plants of the genus. Kirchner et al., for instance, assessed the essential oil extracted from fresh leaves of *Hedyomum brasiliense* Miq. for its antibacterial properties. The study demonstrated no activity against the Gram-negative bacteria, *Escherichia coli* and *Pseudomonas aeruginosa,* but the EO was described as an antibacterial agent against Gram-positive microorganisms, *Staphylococcus aureus, Staphylococcus saprophyticus* and *Bacillus subtilis,* with a MIC value of 0.312% (*v/v*). This research, along with more recent studies provided by Murakami et al., also confirmed the antifungal properties of *H. brasiliense* against the dermatophytes *M. canis*, *M. gypseum*, *T. mentagrophytes*, and *T. rubrum*, and the yeasts *Candida albicans* and *C. parapsilosis* [12,15]. Another investigation into the genus is a comparison of male and female specimens of *H. racemosum* Ruiz & Pav. by Valarezo et al. The essential oil extracted from their leaves also exhibited low activity against pathogenic bacteria and fungi. As opposed to the other species described, female plants proved to be effective against *Klebsiella pneumoniae*, which is a Gram-negative bacterium [16].

Several EOs from known aromatic species like Lavender, Thyme or Peppermint have demonstrated the antimicrobial potential of volatile components occurring in plants. Particularly, thymol and *γ*-terpinene have shown good antifungal profiles, being *p*-thymol the most prominent compound, interfering at the cell wall level, modifying the permeability of the lipidic bilayer of fungal strains [17]. Despite being the antiviral effect the most cited effect for EOs, the antimicrobial effect displayed by several aromatic species, including *Hedyosmum,* encouraged us to value the antimicrobial potential of these two related species.

The *H. spectabile* EO proved a null or deficient scavenging effect for DPPH, with an SC_50_ of 2366.6 ± 2.99 µg/mL and moderate activity for ABTS, exerting an SC_50_ of 214.41 ± 4.03 µM; meanwhile, the reported antioxidant activity for *H. cumbalense* EO was greater, with an SC_50_ of 209.99 ± 1.33 µg/mL on the DPPH assay, and a good scavenging effect on the ABTS radical (SC_50_ of 96.02 ± 0.33 µM). It was discovered that the *Hedyosmum* species generally exhibit strong antioxidant activity by comparing it to previous studies on the genus. For instance, Guerrini et al. examined the antioxidant activity of *Hedyosmum sprucei* Solms-Laub. essential oil isolated from aerial portions of the plant and found that it had an IC_50_ value of 230 ± 10 μg/mL for DPPH scavenging [18].

Compared to the other biological activities assessed in this study, the anticholinesterase activity of both essential oils was the most notable. *H. cumbalense* demonstrated strong inhibitory activity against AChE with an IC_50_ value of 61.94 ± 1.04 µg/mL, while *H. spectabile* displayed an even greater effect with a value of 21.15 ± 1.03 µg/mL. It is suggested that its primary constituents may be responsible for the potent anticholinesterase activity. In particular, the single compound acyphypllene, found in *H. spectabile* (5.37 ± 0.27%), was identified by Lobato et al. to be a promising metabolite that showed signs of inhibiting acetylcholinesterase receptors [19]. According to Bonesi et al., the compound sabinene, which is found in both essential oils and is a prominent component of *H. cumbalense* (14.37 ± 1.64%), also exhibited high activity against AChE with an IC_50_ value of 176.5 µg/mL [20].

It is of vital importance when assessing anticholinesterase effects, which are used to find anti-Alzheimer’s compounds, to asses also the antioxidant potential of substances present in aromatic species. Our research group, besides investigating the antimicrobial potential of EOs, also determined their potential as possible chemopreventive Alzheimer’s agents. In order to determine this, antiradical assays with inhibitory in vitro studies of acetylcholinesterase were conducted.

It is well known the relationship between inflammation and Alzheimer’s progress in patients and also that the inflammation mostly occurs by a disbalance between oxidants and antioxidants. In Alzheimer’s disease, the activity of antioxidant enzymes is reduced; for that reason, the presence of endogenous or exogenous antioxidants is vital because they prevent or slow the damage of cells caused by free radicals. According to Sinyor et al., while in vitro or in vivo studies have demonstrated that antioxidants rich nutrients can protect the brain from oxidative damage, there are limited data about epidemiological and clinical trial studies; however, the evidence suggests that antioxidants could reduce the risk of inflammatory events that lead to AD progress [21].

Our research group has performed extensive research about the biological properties of essential oils of aromatic species from southern Ecuador, but at the moment, we have investigated only *H. racemosum male and female specimens and H. strigosum* EO and found that the antimicrobial activity is relevant only in the case of *H. strigosum* were it was demonstrated a strong antimicrobial effect over *Campylobacter jejuni* and two dermatophytes fungi and again, thymol, was found to be the most abundant compound [16,22] which according to literature present antifungal properties.

## 4. Materials and Methods

### 4.1. Plant Material

*Hedyosmum cumbalense* and *Hedyosmum spectabile* leaves were collected under permission granted by Ministerio del Ambiente de Ecuador (Ecuadorian Environmental Ministry) by means of authorization No. MAE-DNB-CM-2016-0048. The leaves of *Hedyosmum cumbalense* were collected in the month of November in the locality of Cerro Toledo, canton Loja, province of Loja, Ecuador, at 3100 m a.s.l. at a latitude of 4°22′31″ S and longitude of 79°06′39″ W. Vouchers have been deposited at the herbarium of the Universidad Técnica Particular de Loja (UTPL), with accession code HUTPL 14548. 

The leaves of *Hedyosmum spectabile* were collected in the month of October in the locality El Tiro, canton Loja, province of Loja, Ecuador, at 2850 m a.s.l. at a latitude of 3°59′28″ S and a longitude of 79°08′56″ W. Vouchers have been deposited at the herbarium of the Universidad Técnica Particular de Loja (UTPL), with accession code HUTPL 14547.

### 4.2. Postharvest Treatments

Both plants were processed upon their arrival at the laboratory: the leaves were separated from foreign material. The leaves of *H. spectabile* were dried in an incubator at 25 °C for 24 h. The vegetal material obtained from *H. cumbalense* was analyzed 1 or 2 h after it arrived at the laboratory.

### 4.3. Essential Oil Extraction

The method used for the extraction of the essential oil was steam distillation, performed in a Clevenger-type apparatus for 4 h. After the procedure, the 2 remaining layers, corresponding to water on the bottom and the oil on the top, were separated using a pipette. The oil extracted was stored in an amber vial at 4 °C to prevent the loss of its content and the alteration of the compounds. This method was used for both plants.

### 4.4. Identification of the Composition of the Essential Oils

#### 4.4.1. Qualitative Analysis

Both samples were diluted (1/100, *v/v*, EO/DCM) and analyzed using a Thermo Scientific Gas Chromatograph (TRACE 1300 Series) coupled to a Single Quadrupole Mass Spectrometer (ISQ 7000 Series). A non-polar DB-5 ms column of 0.25 mm × 30 m, a thickness of 0.25 µm (5%-phenyl-methylpolyxilosane) was used. For each run, 1 µL of the EO dilutions were injected with a split ratio of 1:40. The equipment operated with electronic ionization (70 eV). Methane was used as a carrier gas at 1 mL/min in constant flow mode. The initial oven temperature was 60 °C, following a gradient until 230 °C was reached. Each run lasted 58 min. The Retention Index was calculated using alkanes from C9 to C10, which were previously injected under the same conditions. The compounds were identified through a comparison between the CRI and reference literature [23].

#### 4.4.2. Quantitative Analysis

An Agilent Gas Chromatograph (model 6890N series) equipped with a flame ionization (FID) detector was used. The GC-FID analyses were performed using the same method and instrumental configuration as GC-MS, which was previously described. The run time was 58 min.

#### 4.4.3. Enantioselective Analysis

For the enantioselective analysis, a Thermo Scientific Gas Chromatograph (TRACE 1300 Series) using a chiral 2,3-diethyl-6-tert-butyldimethylsilyl-β-cyclodextrin-based column was used. The running conditions were similar to those applied for the qualitative and quantitative analyses. The run time was 88 min.

### 4.5. Antimicrobial Activity

Using the method outlined by Valarezo et al., the antibacterial activity of *H. cumbalense* and *H. spectabile* essential oils were evaluated against three Gram-positive bacteria (*Enterococcus faecalis* ATCC ^®^ 19433, *Enterococcus faecium* ATCC ^®^ 27270 and *Staphylococcus aureus* ATCC ^®^ 25923), two Gram-negative bacteria (*Escherichia coli* (O157:H7) ATCC ^®^ 43888 and *Pseudomonas aeruginosa* ATCC ^®^ 10145), and two yeasts and sporulated fungi (*Candida albicans* ATTC ^®^ 10231 and *Aspergillus niger* ATCC ^®^ 6275) (Table 4). The essential oil was dissolved in DMSO, and the bacterial strains were cultured in Müeller-Hinton (MH) broth. Ampicillin (1 g/mL), Ciprofloxacin (1 mg/mL), and Amphotericin B (250 g/mL) were employed as positive controls for Gram-positive cocci, Gram-negative bacilli, and yeasts and sporulated fungi, respectively. Minimum inhibitory concentration (MIC) findings were obtained using DMSO as a negative control (the lowest concentration of the sample capable of inhibiting all visible signs of growth of the microorganism) [24].

### 4.6. Antioxidant Capacity

#### 4.6.1. The 2,2-Diphenyl-1-picril hydrazyl (DPPH) Radical Scavenging Assay

The methodology presented by Salinas et al. [25], which summarizes the DPPH radical scavenging assay put out by Thaipong et al. [26] with just minor changes, was used to evaluate DPPH. The 2,2-diphenyl-1-picrylhydryl free radical (DPPH-) was used. For the creation of the solution used, 24 mg of DPPH was dissolved in 100 mL of methanol. This solution was stabilized at 515 nm in an EPOCH 2 microplate reader until an absorbance of 1.1 ± 0.01 was attained. For the antiradical reaction, different concentrations of the essential oil were used (1, 0.5, and 0.25 mg/mL). Then, 30 mL of the EO sample and 270 mL of the DPPH-adjusted working solution were each added to a 96-microwell plate. The conditions in which the reaction was observed were 515 nm for 1 hour at room temperature. The values of SC_50_ were computed according to the curve generated from previous data, in which methanol was used as blank. Trolox was employed as a positive control. The essential oils of *H. cumbalense* and *H. spectabile* were both analyzed using the same process.

#### 4.6.2. The 2,2-Azino-bis(3-ethylbenzothiazoline-6-sulfonic acid) (ABTS) Radical Scavenging Assay

The procedure used was based on the method employed to estimate the antioxidant power evaluated against the ABTS•+ cation (2,2′-azinobis-3-ethylbenzothiazoline-6-sulfonic acid) described by Salinas et al. [25]. The technique was created using the information provided by Arnao et al. [27] and Thaipong et al. [26], with a few minor alterations. As a first step, a stock solution of the radical was made by stirring equal volumes of potassium persulfate (2.6 M) and ABTS (7.4 M) for 12 h. For the creation of the standard solution, it was dissolved in methanol until an absorbance of 1.1 ± 0.02 at 734 nm was reached using an EPOCH 2 microplate reader. The antiradical response was examined over the course of 60 min at room temperature. It was done by plating 270 mL of the ABTS working adjusted solution and 30 mL of the essential oils at various doses (1, 0.5, and 0.25 mg/mL). The blank, positive control, and calculations were the same as the DPPH assay.

### 4.7. Anticholinesterase Assay

The anticholinesterase assay was based on the procedure presented by Andrade et al. [28], in which they followed the methodology proposed by Ellman et al. [29], with minor modifications as recommended by Rhee et al. [30]. For the reaction mixture, 40 μL of Buffer Tris were incorporated into 20 μL of the tested sample solution, along with 20 μL of acetylthiocholine (ATCh, 15 mM, PBS pH 7.4), and 100 L of DTNB (3 Mm, Buffer Tris). Then, the pre-incubation was executed for 3 min at room temperature, under constant shaking. The reaction was then begun by adding 20 μL of 0.5 U/mL AChE. The quantity of product produced was measured using an EPOCH 2 microplate reader at 405 nm for an hour at room temperature. Ten milligrams of essential oil were dissolved in MeOH to create EO sample solutions. To reach final concentrations of 1000, 100, and 10 g/mL, three more 10-factor dilutions were added. The corresponding curve fitting of the data received from the computed rate of reactions was used to get the IC50 value. The calculated IC_50_ value for donepezil-hydrochloride, which served as a positive control, was 12.40 ± 1.35 nM. The *H. cumbalense* and *H. spectabile* essential oils were assessed under the same methodology.

## Figures and Tables

**Figure 1 plants-12-00039-f001:**
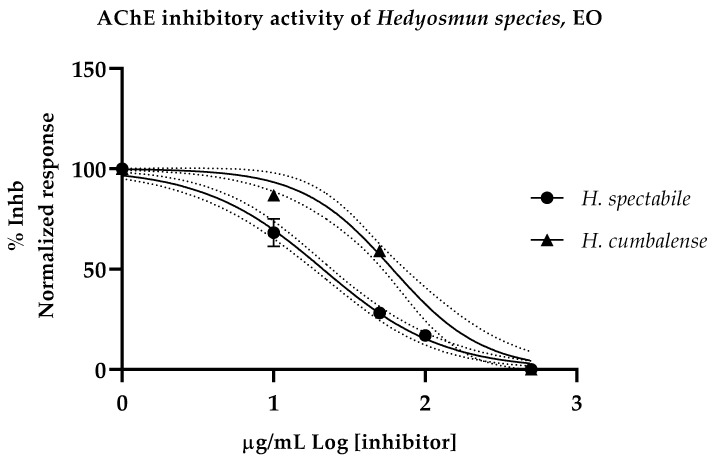
Inhibitory effect graph of *H. spectabile* and *H. cumbalense*, EO, against acetylcholinesterase. Data were built from three replicas of three different experiments and analyzed by the non-linear regression model.

**Table 1 plants-12-00039-t001:** Chemical compounds present in the essential oil of the leaves of *H. cumbalense*.

Nº	Compounds	LRI^a^	LRI^b^	%	CF
1	*α*-Thujene	920	924	0.32 ± 0.08	C_10_H_16_
2	*α*-Pinene	927	932	7.91 ± 2.10	C_10_H_16_
3	Camphene	944	946	3.01 ± 0.72	C_10_H_16_
4	Sabinene	968	969	14.37 ± 1.64	C_10_H_16_
5	*β*-Pinene	973	974	7.41 ± 1.46	C_10_H_16_
6	Myrcene	985	988	0.56 ± 0.07	C_10_H_16_
7	*α*-Phellandrene	1004	1002	0.52 ± 0.04	C_10_H_16_
8	*α*-Terpinene	1014	1014	0.65 ± 0.06	C_10_H_16_
9	*ρ*-Cymene	1023	1020	0.93 ± 0.02	C_10_H_14_
10	Sylvestrene	1026	1025	2.07 ± 0.17	C_10_H_16_
11	*β*-Phellandrene	1028	1025	0.18 ± 0.01	C_10_H_16_
12	1,8-Cineole	1029	1026	1.89 ± 0.48	C_10_H_18_O
13	(Z)-*β*-Ocimene	1032	1032	1.30 ± 0.08	C_10_H_16_
14	(E)-*β*-Ocimene	1042	1044	0.64 ± 0.05	C_10_H_16_
15	*γ*-Terpinene	1054	1054	1.23 ± 0.12	C_10_H_16_
16	cis-Sabinene hydrate (IPP vs. OH)	1069	1065	0.59 ± 0.15	C_10_H_18_O
17	Terpinolene	1082	1086	0.37 ± 0.04	C_10_H_16_
18	6-Camphenone	1091	1095	0.91 ± 0.17	C_10_H_14_O
19	Linalool	1101	1095	4.94 ± 0.93	C_10_H_18_O
20	trans-Sabinene hydrate (IPP vs. OH)	1103	1098	0.31 ± 0.11	C_10_H_18_O
21	Chrysanthenone	1124	1124	0.22 ± 0.05	C_10_H_14_O
22	cis-*ρ*-Menth-2-en-1-ol	1125	1118	0.32 ± 0.11	C_10_H_18_O
23	*α*-Campholenal	1128	1122	0.42 ± 0.05	C_10_H_16_O
24	trans-Pinocarveol	1141	1135	0.30 ± 0.05	C_10_H_16_O
25	trans-Verbenol	1147	1140	1.18 ± 0.31	C_10_H_16_O
26	Camphor	1149	1141	1.17 ± 0.39	C_10_H_16_O
27	Citronellal	1153	1148	3.51 ± 0.37	C_10_H_18_O
28	Eucarvone	1157	1146	0.17 ± 0.04	C_10_H_14_O
29	cis-Chrysanthenol	1160	1160	0.43 ± 0.11	C_10_H_16_O
30	Pinocarvone	1164	1160	1.28 ± 0.32	C_10_H_14_O
31	Borneol	1173	1165	0.93 ± 0.26	C_10_H_18_O
32	cis-Pinocamphone	1178	1172	0.56 ± 0.20	C_10_H_16_O
33	Terpinen-4-ol	1181	1174	2.43 ± 0.86	C_10_H_18_O
34	cis-Dihydro carvone	1186	1191	0.33 ± 0.07	C_10_H_16_O
35	neo-Dihydro carveol	1198	1193	1.50 ± 0.38	C_10_H_18_O
36	trans-Piperitol	1212	1207	0.32 ± 0.17	C_10_H_18_O
37	Thymol, methyl ether	1231	1232	4.75 ± 0.66	C_11_H_16_O
38	Neral	1243	1235	0.99 ± 0.20	C_10_H_16_O
39	Geraniol	1255	1249	0.23 ± 0.07	C_10_H_18_O
40	Geranial	1273	1264	1.75 ± 0.24	C_10_H_16_O
41	Isobornyl acetate	1283	1283	9.12 ± 1.11	C_12_H_20_O_2_
42	*ρ*-Cymen-7-ol	1297	1289	0.15 ± 0.09	C_10_H_14_O
43	*α*-Terpinyl acetate	1346	1346	0.67 ± 0.09	C_12_H_20_O_2_
44	Citronellyl acetate	1350	1350	0.68 ± 0.08	C_12_H_22_O_2_
45	Eugenol	1357	1356	2.55 ± 0.99	C_10_H_12_O_2_
46	Neryl acetate	1359	1359	0.16 ± 0.04	C_12_H_20_O_2_
47	Geranyl acetate	1379	1379	1.95 ± 0.17	C_12_H_20_O_2_
48	Cyperene	1394	1398	0.20 ± 0.08	C_15_H_24_
49	Methyl eugenol	1406	1403	0.50 ± 0.14	C_11_H_14_O_2_
50	(E)-Caryophyllene	1412	1417	0.19 ± 0.04	C_15_H_24_
51	*β*-Copaene	1422	1430	0.23 ± 0.05	C_15_H_24_
52	Rotundene	1455	1457	0.15 ± 0.05	C_15_H_24_
53	allo-Aromadendrene	1466	1458	0.23 ± 0.07	C_15_H_24_
54	Germacrene D	1475	1480	0.70 ± 0.28	C_15_H_24_
55	*α*-Selinene	1494	1498	0.19 ± 0.06	C_15_H_24_
56	*δ*-Amorphene	1512	1511	0.25 ± 0.15	C_15_H_24_
57	Eugenol acetate	1520	1521	1.60 ± 0.33	C_12_H_14_O_3_
58	Germacrene B	1554	1559	0.23 ± 0.04	C_15_H_24_
59	Spathulenol	1574	1577	0.71 ± 0.19	C_15_H_24_O
60	Caryophyllene oxide	1580	1582	0.67 ± 0.22	C_15_H_24_O
61	Globulol	1584	1590	0.12 ± 0.02	C_15_H_26_O
62	Carotol	1600	1594	0.81 ± 0.21	C_15_H_26_O
63	cis-Isolongifolanone	1611	1612	0.33 ± 0.09	C_15_H_24_O
64	Silphiperfol-6-en-5-one	1623	1624	0.15 ± 0.04	C_15_H_22_O
65	Daucol	1646	1641	0.31 ± 0.14	C_15_H_26_O_2_
66	Elemol acetate	1679	1680	0.16 ± 0.07	C_17_H_28_O_2_
	HM			21.80	
	OM			37.79	
	HS			13.08	
	OS			10.18	
	Others			13.08	
	Total			95.94	

Percentage (%) is expressed as mean ± SD (standard deviation); LRI^a^: Calculated Linear Retention Index; LRI^b^: Linear Retention Index read in the bibliography; CF: Condensed formula. HM: hydrocarbonated monoterpenes; OM: oxygenated monoterpenes; HS: hydrocarbonated sesquiterpenes; OS: oxygenated sesquiterpenes.

**Table 2 plants-12-00039-t002:** Chemical compounds present in the essential oil of the leaves of *H*. *spectabile*.

Nº	Compounds	LRI^a^	LRI^b^	%	CF
1	*α*-Pinene	928	932	1.64 ± 0.51	C_10_H_16_
2	Sabinene	969	969	0.41 ± 0.20	C_10_H_16_
3	*β*-Pinene	975	974	0.25 ± 0.12	C_10_H_16_
4	Myrcene	987	988	0.24 ± 0.04	C_10_H_16_
5	Sylvestrene	1027	1025	0.29 ± 0.15	C_10_H_16_
6	(Z)-*β*-Ocimene	1034	1032	1.84 ± 0.53	C_10_H_16_
7	(E)-*β*-Ocimene	1044	1044	5.35 ± 1.01	C_10_H_16_
8	Linalool	1102	1095	0.99 ± 0.53	C_10_H_18_O
9	Chrysanthenone	1123	1124	0.68 ± 0.35	C_10_H_14_O
10	1,3,8-*ρ*-Menthatriene	1131	1108	4.99 ± 1.15	C_10_H_18_O
11	Verbenone	1200	1204	0.31 ± 0.23	C_10_H_14_O
12	Bornyl acetate	1285	1284	0.25 ± 0.16	C_12_H_20_O_2_
13	Myrtenyl acetate	1326	1324	0.84 ± 0.28	C_12_H_18_O_2_
14	*δ*-Elemene	1331	1335	0.88 ± 0.14	C_15_H_24_
15	*α*-Cubebene	1343	1348	0.35 ± 0.26	C_15_H_24_
16	*α*-Copaene	1371	1374	2.48 ± 0.46	C_15_H_24_
17	*β*-Cubebene	1384	1387	0.4 ± 0.17	C_15_H_24_
18	*β*-Elemene	1386	1389	3.69 ± 0.87	C_15_H_24_
19	(E)-Caryophyllene	1414	1417	1.04 ± 0.26	C_15_H_24_
20	*β*-Gurjunene	1426	1431	0.15 ± 0.03	C_15_H_24_
21	6,9-Guaiadiene	1438	1442	0.17 ± 0.06	C_15_H_24_
22	*α*-Humulene	1452	1452	0.74 ± 0.23	C_15_H_24_
23	*α*-neo-Clovene	1456	1452	0.53 ± 0.10	C_15_H_24_
24	Dauca-5,8-diene	1469	1471	0.35 ± 0.12	C_15_H_24_
25	*γ*-Muurolene	1473	1478	1.94 ± 0.49	C_15_H_24_
26	cis-Muurola-4(14),5-diene	1479	1479	17.87 ± 2.79	C_15_H_24_O
27	*δ*-Selinene	1484	1492	0.98 ± 0.25	C_15_H_24_
28	*β*-Selinene	1486	1489	1.69 ± 0.95	C_15_H_24_
29	trans-Muurola-4(14),5-diene	1489	1493	0.61 ± 0.14	C_15_H_24_
30	*α*-Zingiberene	1493	1493	1.84 ± 0.35	C_15_H_24_
31	cis-*β*-Guaiene	1496	1492	0.54 ± 0.10	C_15_H_24_
32	Valencene	1499	1496	1.54 ± 0.37	C_15_H_24_
33	Aciphyllene	1504	1501	5.37 ± 0.27	C_15_H_24_
34	(Z)-*α*-Bisabolene	1508	1506	0.98 ± 0.44	C_15_H_24_
35	Cubebol	1514	1514	0.7 ± 0.04	C_15_H_26_O
36	*δ*-Amorphene	1516	1511	2.44 ± 0.28	C_15_H_24_
37	*γ*-Patchoulene	1521	1502	1.02 ± 0.46	C_15_H_24_
38	trans-Cadina-1,4-diene	1531	1533	0.19 ± 0.10	C_15_H_24_
39	*α*-Cadinene	1536	1537	0.24 ± 0.11	C_15_H_24_
40	*α*-Copaen-11-ol	1544	1539	1.39 ± 0.14	C_15_H_24_O
41	Germacrene B	1558	1559	0.94 ± 0.07	C_15_H_24_
42	(E)-Nerolidol	1561	1561	1.18 ± 0.28	C_15_H_26_O
43	Spathulenol	1579	1577	1.34 ± 0.28	C_15_H_24_O
44	Globulol	1596	1590	2.03 ± 0.57	C_15_H_26_O
45	*β*-Oplopenone	1606	1607	0.19 ± 0.10	C_15_H_24_O
46	Guaiol	1610	1600	0.20 ± 0.09	C_15_H_26_O
47	1,10-di-epi-Cubenol	1613	1618	0.65 ± 0.14	C_15_H_26_O
48	Isolongifolan-7-*α*-ol	1621	1618	0.2 ± 0.07	C_15_H_26_O
49	*γ*-Eudesmol	1626	1630	1.44 ± 0.33	C_15_H_26_O
50	Muurola-4,10(14)-dien-1-*β*-ol	1635	1630	6.55 ± 0.58	C_15_H_24_O
51	Cubenol	1646	1645	2.77 ± 0.22	C_15_H_26_O
52	Valerianol	1654	1656	0.42 ± 0.08	C_15_H_26_O
53	*α*-Cadinol	1662	1652	2.12 ± 0.24	C_15_H_26_O
54	neo-Intermedeol	1666	1658	0.79 ± 0.34	C_15_H_26_O
55	Germacra-4(15),5,10(14)-trien-1-*α*-ol	1691	1685	2.09 ± 0.50	C_15_H_24_O
56	Eudesma-4(15),7-dien-1-*β*-ol (impure)	1693	1687	0.34 ± 0.19	C_15_H_24_O
57	Amorpha-4,9-dien-2-ol	1697	1700	0.90 ± 0.23	C_15_H_24_O
	HM			11.34	
	OM			6.48	
	HS			40.51	
	OS			30.79	
	Others			3.24	
	Total			92.37	

Percentage (%) is expressed as mean ± SD (standard deviation); LRI^a^: Calculated Linear Retention Index; LRI^b^: Linear Retention Index read in the bibliography; CF: Condensed formula. HM: hydrocarbonated monoterpenes; OM: oxygenated monoterpenes; HS: hydrocarbonated sesquiterpenes; OS: oxygenated sesquiterpenes.

**Table 3 plants-12-00039-t003:** Enantioselective analysis of the EOs with a 2,3-diethyl-6-tert-butyldimethylsilyl-β-cyclodextrin-based column.

Enantiomers	LRI	Enantiomeric Ratio (%)	Enantiomeric Excess (%)
*H. cumbalense*			
(1R,5R)-(+)-sabinene	999	34.51	30.98
(1S,5S)-(–)-sabinene	1000	65.49	
(S)-(+)-terpinen-4-ol	1270	21.27	57.47
(R)-(−)-terpinen-4-ol	1275	78.73	
*H. spectabile*			
(1R,5R)-(+)-sabinene	999	47.87	4.25
(1S,5S)-(–)-sabinene	1001	52.13	
(S)-(+)-linalool	1198	0.13	99.74
(R)-(-)-linalool	1202	99.87	

LRI = Calculated linear retention indices.

**Table 4 plants-12-00039-t004:** The antibacterial capacity of essential oils of *H. cumbalense* and *H. spectabile* against pathogenic reference strains measured as the minimum inhibitory concentration (MIC) and expressed in µg/mL.

Microorganism	*H. cumbalense*	*H. spectabile*	Positive Control *
	(µg/mL)	(µg/mL)	(µg/mL)
**Gram-positive cocci**			
Enterococcus faecalis ATCC ^®^ 19433	-	-	0.78
Enterococcus faecium ATCC ^®^ 27270	4000	-	<0.39
Staphylococcus aureus ATCC ^®^ 25923	4000	-	<0.39
**Gram-negative bacilli**			
Escherichia coli (O157:H7) ATCC ^®^ 43888	-	-	1.56
Pseudomonas aeruginosa ATCC ^®^ 10145	-	-	<0.39
**Yeasts and sporulated fungi**			
Candida albicans ATTC ^®^ 10231	1000	2000	<0.09
Aspergillus niger ATCC ^®^ 6275	1000	1000	<0.09

* Ampicillin (1 g/mL) for Gram-positive cocci, Ciprofloxacin (1 mg/mL) for Gram-negative bacilli and Amphotericin B (250 µg/mL) for yeasts and sporulated fungi. (-) Non-active at the highest dose tested 4000 ug/mL.

**Table 5 plants-12-00039-t005:** *H. cumbalense* and *H. spectabile* essential oils antioxidant activity.

Sample	DPPH	ABTS
	SC_50_ (µg/mL—µM *) ± SD	
*H. spectabile*	2366.6 ± 2.99	214. 41 ± 4.03
*H. cumbalense*	209.99 ± 1.33	96.02 ± 0.33
Trolox	29.99 ± 1.04	23.27 ± 1.05

* SC50: Half scavenging capacity expressed as µM for Trolox.

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
