# Peer review of "Chemical Profiling of Hedyosmum cumbalense and Hedyosmum spectabile (Chloranthaceae) Essential Oils, and Their Antimicrobial, Antioxidant, and Anticholinesterase Properties"

_plants, 2022, doi:10.3390/plants12010039_

Round 1

Reviewer 1 Report

This is a well-written manuscript on the antimicrobial properties of essential oils isolated from Hedyosmum cumbalense and Hedyosmum spectabile. I don't have any major concerns with the article in its present form. I suggest the authors explain better why they tested the antioxidants, antimicrobial and anticholinesterase properties of the essential oils analysed here. This is a random collection of tests; therefore, a detailed explanation of what motivated the authors to do these analyses would be very welcome. 

Author Response

All the authors thnak for the suggestion made by the reviewer and all the answers have been included in the main text accordingly. Attached you cand find a document with the changes made.

Reviewer 2 Report

The paper is quite interesting and well written. Some minor points.

According to Plants of the world on line (https://powo.science.kew.org/taxon/urn:lsid:ipni.org:names:7058-1), that should be also cited, 45 taxa have the rang of accepted species. Authors should insert the botanical authority as well as the synonyms.

The Greek symbols of the compound names should be written in italics.

line 128. H. spectabile italics

line 130. H. cumbalense italics

In 3. Discussion authors should insert more details on the previous studied essential oils of Hedyosmum species, highlighting the difference and similarity with respect to their results.

Author Response

All the authors thank for the suggestion made by the reviewer and all the answers have been included in the main text accordingly. Attached you cand find a document with the changes made.
